# Cardiology involvement and mortality in adult patients with advanced solid cancer complicated by atrial fibrillation

**Takeshi Sato**[1,2,◉], **Zhehao Dai** [3,4,*,◉], **Jun Hashimoto**[5,*], **Sachiko Ohde**[6], **Nobuyuki Komiyama**[4], **Takayuki Inomata**[2], **Teruo Yamauchi**[7,8]

1 Department of Cardiology, Nagaoka Chuo General Hospital, Niigata, Japan, 2 Department of Cardiovascular Medicine, Niigata University Graduate School of Medical and Dental Sciences, Niigata, Japan, 3 Department of Cardiovascular Medicine, The University of Tokyo Graduate School of Medicine, Tokyo, Japan, 4 Department of Cardiovascular Medicine, St. Luke's International Hospital, Tokyo, Japan, 5 Department of Medical Oncology, St. Luke's International Hospital, Tokyo, Japan, 6 St. Luke's International University Graduate School of Public Health, Tokyo, Japan, 7 Cancer Biology Program, Translational and Clinical Research, University of Hawaiʻi Cancer Center, Honolulu, Hawaii, USA, 8 Department of Oncology, The Queen's Medical Center, Honolulu, Hawaii, USA

◉ These authors contributed equally to this work.
* daizh@luke.ac.jp (ZD); juhashim@luke.ac.jp (JH)

## Abstract

### Background

The association between comorbid atrial fibrillation (AF) and survival in adult patients with advanced solid cancer, as well as the impact of cardiology involvement in such patients, remains unclear.

### Methods

This retrospective cohort study included adult patients diagnosed with advanced solid cancers. We calculated prevalence of AF in different cancer types and compared all-cause mortality between patients with and without AF. We further examined the association between cardiology involvement and mortality in the subset of participants with AF.

### Results

Among the 1,349 adult patients with advanced solid cancer, 122 (9.0%) had AF. The risk of AF was the highest in lung and mediastinal cancer (15.6%). AF was associated with higher all-cause mortality, which became neutral after adjustment for age, sex, comorbidities, cancer types and cancer treatments (crude hazard ratio [HR] 1.39, 95% confidence interval [CI] 1.11–1.75, p = 0.004; adjusted HR 1.08, 95%CI 0.84–1.39, p = 0.552). In those with AF, cardiology involvement was independently associated with lower all-cause mortality (age, sex, comorbidities, cancer types and cancer treatments-adjusted HR 0.50 [95%CI 0.28–0.88], p = 0.017), though the cumulative incidence of neither cardiovascular nor non-cardiovascular death differed significantly between patients who received cardiology care and those who did not.

**Data availability statement:** All relevant data are within the manuscript and its Supporting information files.

**Funding:** The author(s) received no specific funding for this work.

**Competing interests:** The authors have declared that no competing interests exist.

## Conclusions

In adult patients with advanced solid cancer, AF *per se* was not independently associated with increased mortality. Cardiology involvement in patients with advanced solid cancer and AF was linked to a better overall survival, but with low certainty that this finding is not attributable to unmeasured confounding.

## Introduction

With recent advancements in cancer screening, diagnosis, and treatment, prognosis has improved substantially [1]. As survival is prolonged, late morbidity and mortality have become major concerns in managing patients with cancer. Cardiovascular disease is the second most frequent cause of late morbidity and mortality among cancer survivors [2,3], represented prominently by atrial fibrillation (AF). Patients with cancer are at a 1.5 to 3-fold higher risk of developing AF than in general population [4,5]. Furthermore, among patients with cancer, those with AF have a two-fold increased risk of ischemic stroke and a six-fold increased risk of heart failure compared to those without AF [3]. A previous study based on the health insurance database in Taiwan showed that new-onset AF in patients with cancer is associated with an increased risk of thromboembolism and heart failure, whereas baseline AF is linked to increased mortality [6]. Another study found that newly developed AF during the postoperative period is associated with higher mortality in patients with esophageal cancer who undergo esophagectomy [7]. Further research is needed to explore the association between AF and mortality in patients with cancer, as well as to identify interventions that could potentially reduce mortality in this population.

Initiating anticoagulation in patients with AF and cancer is more complicated than in the general AF population. Patients with cancer face higher risks of both thromboembolism and bleeding [8]. Conventional risk assessment tools for thromboembolism and bleeding are not directly applicable to patients with cancer due to their distinct risk profiles [9,10]. The 2022 European Society of Cardiology Guidelines on cardio-oncology recommend a comprehensive evaluation of thromboembolic risk, bleeding risk, and drug-drug interactions for these patients to determine anticoagulant use (Class of recommendation I, Level of evidence C) [11]. This approach may be promoted by a multidisciplinary collaboration between oncologists and cardiologists. A recent report demonstrated that although patients with AF and cancer were less likely to receive care from cardiologists than those with AF but not cancer (54% vs. 62%), cardiology involvement in this population was associated with increased anticoagulant prescriptions and reduced stroke risk [12]. Furthermore, both cancer and AF increase the risk of heart failure [3,11]. Therefore, cardiology involvement in managing these multimorbid conditions is expected to reduce morbidity and mortality. However, whether cardiology involvement lowers mortality in patients with cancer complicated by AF, particularly those with advanced cancer, remains unclear.

We hypothesized that AF contributed to higher mortality in patients with advanced solid cancer due to increased risks of heart failure and ischemic stroke [3], which might be mitigated by cardiology involvement with appropriate approaches to anticoagulation and anti-arrhythmic drugs, and heart failure therapies. Thus, this single-center retrospective cohort study, including patients with advanced solid cancer, was conducted to compare the mortality between patients with and without AF. Moreover, among those with AF, this study also examined differences in mortality between patients who received care from a cardiologist and those who did not.

## Methods

### Study design and population

This single-center retrospective cohort study recruited patients from St. Luke's International Hospital, Tokyo, Japan. This study was approved by the Institutional Review Board of St. Luke's International University (approval No. 18-J013), which waived written informed consent from patients given the observational nature of the study and the anonymity in the analysis. Instead, all participants in this study provided consent under an opt-out policy. Eligible participants were initially screened if diagnosed with advanced solid cancer between January 2008 and December 2017. Advanced solid cancer was defined as a locally advanced and/or metastatic tumor that was not amenable to curative resection. Exclusions included: (1) patients under 20 years old, (2) those who sought a second opinion without subsequent management at our hospital, and (3) those who did not receive cancer treatment for any reason.

### Data collection

Clinical and demographic data were collected from electronic medical records, encompassing baseline demographics (sex and age), comorbidities (hypertension, diabetes mellitus, ischemic stroke, and heart failure), cancer treatment for the index cancer (either ongoing treatment upon inclusion or those received previously in earlier stages), and mortality. The baseline was defined as the time of advanced solid cancer diagnosis. AF, whether symptomatic or asymptomatic, was diagnosed using a surface electrocardiogram (ECG) following current guidelines [3]. Cardiology involvement was defined as the documented regular participation of a cardiologist in the patient's care within medical records or self-reported regular visits to a cardiologist outside the hospital setting. Data collection was conducted by a healthcare information technician and an independent physician through January 4 to January 31, 2019. The authors had access to anonymous data where individual participants could not be identified.

### Measurement of outcomes

This study was designed to investigate the association between AF and mortality among patients with advanced solid cancer, and to explore whether this association was influenced by cardiology involvement. Therefore, the primary endpoint was defined as all-cause mortality. Secondary endpoints included cardiovascular and non-cardiovascular deaths, specifically analyzed if a statistically significant difference was observed in the primary endpoint comparison.

### Statistical analysis

Continuous variables were expressed as median (interquartile range [IQR]) and were compared between the two groups using Mann-Whitney $U$ test. Categorical variables were expressed as numbers (percentages) and were compared using the $\chi^2$ or Fisher's exact test, as appropriate.

We initially compared the baseline characteristics of the patients with and without AF. Subsequently, to assess overall survival in adult patients with advanced solid cancer with and without AF, we performed Kaplan-Meier estimates and conducted Cox proportional hazard modeling to calculate hazard ratios (HRs) and 95% confidence intervals (CIs). The Cox proportional hazard model was adjusted for age, sex, hemoglobin, creatinine, hypertension, diabetes mellitus, stroke, cancer type, and cancer treatment. Similarly, within the AF subgroup, we compared the overall survival between patients cared by a cardiologist and those without cardiology involvement. We employed Kaplan-Meier estimates and Cox proportional hazard models. Model 1 was adjusted for age, sex, AF classification, hemoglobin, creatinine, and

comorbidities. Model 2 was further adjusted for cancer types. In Model 3, additional adjustments were made for treatments specific to AF. For multivariable regressions, variable selections were based on clinical significance with the forced entry procedure. Multicollinearity in the regression coefficients was assessed by means of the variance inflation factor (VIF) with a criterion of VIF <10. Proportional hazard assumptions were assessed by Schoenfeld residuals tests along with visualization of scaled Schoenfeld residual plots. For analyzing secondary endpoints (cardiovascular and non-cardiovascular deaths), we applied the Aalen-Johansen method and Gray's test to address competing risks (non-cardiovascular death and cardiovascular death, respectively) [13,14].

Statistical significance was set at P < 0.05. All analyses were performed using the R statistical software version 4.4.0 (R Foundation for Statistical Computing).

## Results

### Participants and baseline characteristics

We initially screened 2,075 patients diagnosed with advanced solid cancer between January 2008 and December 2017 at St. Luke's International Hospital, Tokyo, Japan. Of these, 197 patients were excluded because they were under the age of 20 or visited the hospital only for a second opinion without subsequent management. Among the remaining 1,878 patients, 529 were excluded for not receiving any anticancer treatments, such as chemotherapy, endocrine therapy, radiotherapy, or surgery. Ultimately, this study included 1,349 adult patients with advanced solid cancer (Fig 1).

The median age of the included patients was 66 years (range 20–99), and 122 (9.0%) patients had AF. Cancer types were categorized into lung and mediastinal, gastrointestinal, urologic, hepato-biliary-pancreatic, and breast cancers. Less common cancers, such as gynecological cancer, thyroid cancer, oral cancer, head and neck cancer, sarcoma, and carcinoma of unknown primary, were grouped as "others." The frequency of AF was the highest among patients with lung and mediastinal cancers (15.6%) and the lowest in those with breast cancer (2.6%) (Fig 2). Baseline characteristics of patients with and without AF are shown in Table 1. Patients with AF were older (median 74 years [IQR 68–80] vs. median 66 years [IQR 55–74], p < 0.001) and had a lower proportion of women (27.0% vs. 54.0%, p < 0.001) than those without AF. Patients with AF were more likely to have hypertension, diabetes, heart failure, and stroke (p < 0.001, p < 0.001, p < 0.001, and p = 0.013, respectively). Regarding cancer type, patients with AF had a lower frequency of breast cancer and a higher frequency of lung and mediastinal cancer (5.7% vs. 21.1%, p < 0.001; 32.0% vs. 17.2%, p < 0.001, respectively). Patients with different cancer types received different cancer treatment expectedly, both in the entire cohort, and in the subset of patients with AF (S1 and S2 Tables).

### AF, clinical characteristics, and mortality in adult patients with advanced solid cancer

The median follow-up duration was 1.8 years. The overall survival curves of patients with and without AF are shown in Fig 3. Patients with AF had significantly worse survival than those without AF (median survival, 1.6 vs. 2.7 years, log-rank p = 0.004). We then performed multivariable Cox proportional hazards analysis adjusted for age, sex, comorbidities, cancer types, and cancer treatments (VIF of covariates ranged from 1.105 to 3.956 with a mean VIF = 1.543; scaled Schoenfeld residual plots shown in S1 Fig). Age higher than 65-year-old and lower hemoglobin level were independently associated with mortality. Mortality also differed among different cancer types, with gastrointestinal, lung and mediastinal, and hepato-biliary-pancreatic cancers associated with higher mortality than urologic cancer. In addition, cancer treatments such as

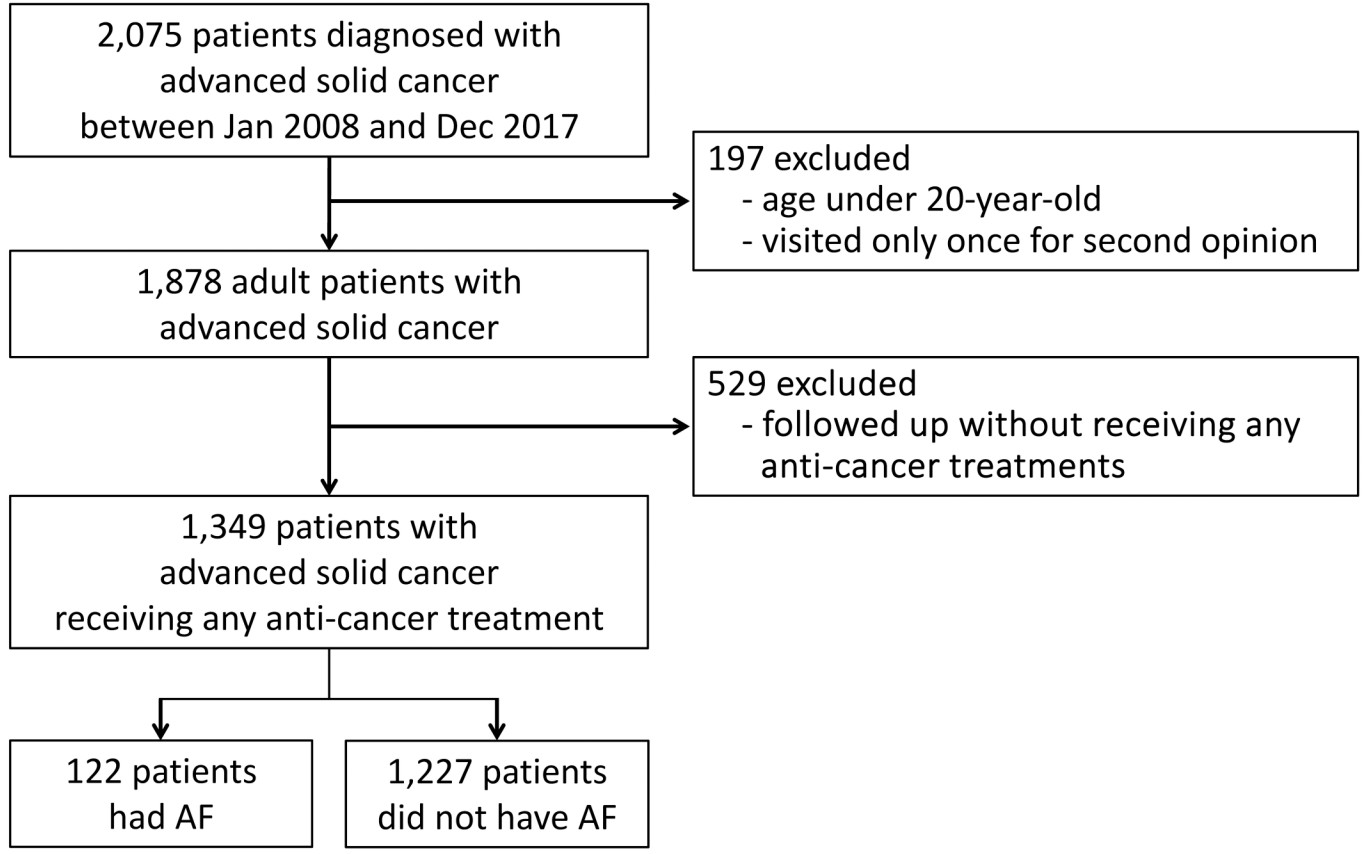

**Fig 1. CONSORT flow diagram of patients inclusion and exclusion.** AF, atrial fibrillation; CONSORT, Consolidated Standards of Reporting Trials.

surgery and endocrine therapy were independently associated with better survival. A positive correlation was observed between the presence of AF and mortality in crude analysis, which became neutral after adjusting for clinical characteristics (adjusted HR 1.08, 95%CI 0.84–1.39, p = 0.552) (Table 2). We further stratified the patients by cancer type, focusing on urologic, gastrointestinal, and lung and mediastinal cancers, which had the largest number of patients. The stratified analysis revealed that, within each cancer type, comorbid AF was not associated with all-cause mortality (S2 Fig).

### Cardiology involvement and all-cause mortality in adult patients with advanced solid cancer and AF

We further divided the patients with AF into two groups: those with cardiology involvement and those without. Paroxysmal and persistent AF were comparably distributed between the two groups. Anticoagulants, beta-blockers, and amiodarone were more frequently administered to patients under cardiology care (Table 3). The overall survival curves are shown in Fig 4, with median survival times of 2.7 years for patients without AF, 2.1 years for those with AF and cardiology involvement, and 1.2 years for those with AF but no cardiology involvement. Significant differences were observed between patients with AF and cardiology involvement and those with AF without cardiology involvement (log-rank p = 0.016), and between patients without AF and those with AF without cardiology involvement (log-rank p < 0.001). Among patients with AF, cardiology involvement was associated with lower all-cause mortality (crude

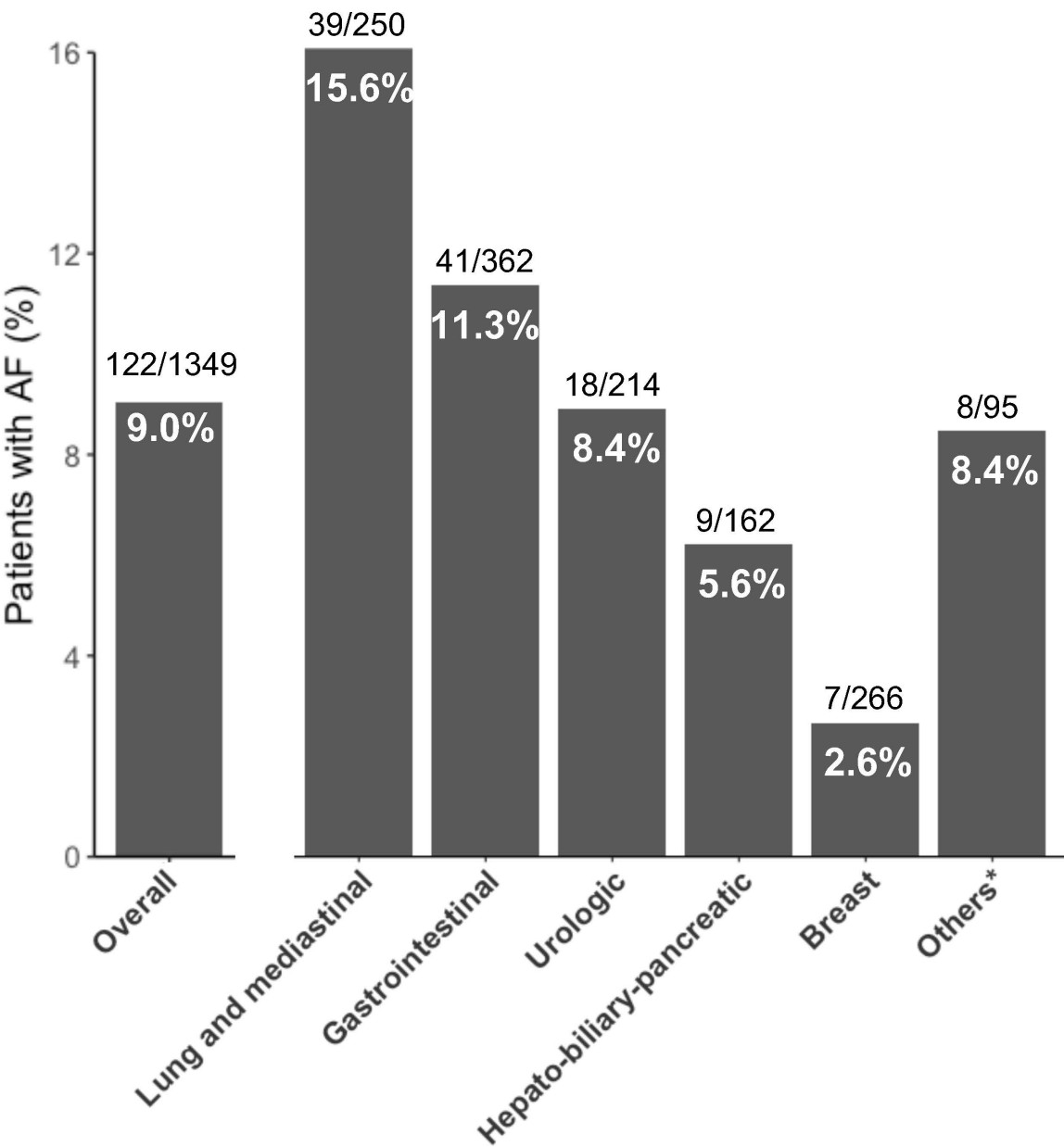

**Fig 2. The risk of AF in adult patients with advanced solid cancer across different cancer types.** *Other cancers included gynecological cancer, thyroid cancer, oral cancer, head and neck cancer, sarcoma, and carcinoma of unknown primary. AF, atrial fibrillation.

HR 0.59, 95%CI 0.38–0.91, p = 0.017), a finding that remained significant after adjusting for comorbidities (model 1), comorbidities and cancer type (model 2), and comorbidities, cancer type, and AF treatments (model 3) (Table 4). VIFs ranged from 1.070 to 1.441 (mean 1.221) for model 1, from 1.140 to 2.399 (mean 1.486) for model 2, and from 1.273 to 3.723 (mean 1.771) for model 3, respectively; and scaled Schoenfeld residual plots were shown in S3 Fig. Additionally, among the 71 patients with AF and cardiology involvement, 66 received any AF treatment (either anticoagulant or antiarrhythmic or both), who presented comparable all-cause mortality with those who did not receive any AF treatment (log-rank p = 0.332).

**Table 1. Baseline characteristics of the participants and comparisons between those with and without AF.**

| | All patients | Without AF | With AF | P-value |
|---|---|---|---|---|
| | (N = 1349) | (N = 1227) | (N = 122) | |
| Age, years | 66 (56–75) | 66 (55–74) | 74 (68–80) | <0.001 |
| Women | 695 (51.5) | 662 (54.0) | 33 (27.0) | <0.001 |
| Hemoglobin (g/dL) | 12.2 (10.6–13.6) | 12.2 (10.6–13.5) | 12.0 (10.1–13.9) | 0.913 |
| Creatinine (mg/dL) | 0.70 (0.57–0.87) | 0.69 (0.56–0.86) | 0.81 (0.68–1.00) | <0.001 |
| Comorbidities | | | | |
| Hypertension | 412 (30.5) | 340 (27.7) | 72 (59.0) | <0.001 |
| Diabetes mellitus | 281 (20.8) | 239 (19.5) | 42 (34.4) | <0.001 |
| Heart failure | 109 (8.1) | 66 (5.4) | 43 (35.2) | <0.001 |
| Stroke | 97 (7.2) | 81 (6.6) | 16 (13.1) | 0.013 |
| Neoplasm type | | | | |
| Gastrointestinal | 362 (26.8) | 321 (26.2) | 41 (33.6) | 0.096 |
| Urologic | 214 (15.9) | 196 (16.0) | 18 (14.8) | 0.824 |
| Hepato-biliary-pancreatic | 162 (12.0) | 153 (12.5) | 9 (7.4) | 0.133 |
| Breast | 266 (19.7) | 259 (21.1) | 7 (5.7) | <0.001 |
| Lung and mediastinal | 250 (18.5) | 211 (17.2) | 39 (32.0) | <0.001 |
| Other cancers* | 95 (7.0) | 87 (7.1) | 8 (6.6) | 0.973 |
| Staging of neoplasm | | | | 0.787† |
| Stage II ~ III | 42 (3.1) | 38 (3.1) | 4 (3.3) | |
| Stage IV | 1307 (96.9) | 1189 (96.9) | 118 (96.7) | |
| Treatment for neoplasm | | | | |
| Surgery | 473 (35.1) | 427 (34.8) | 46 (37.7) | 0.588 |
| Radiotherapy | 155 (11.5) | 136 (11.1) | 19 (15.6) | 0.182 |
| Chemotherapy | 956 (70.9) | 879 (71.6) | 77 (63.1) | 0.061 |
| Endocrine therapy | 274 (20.3) | 256 (20.9) | 18 (14.8) | 0.138 |

Data are presented as median (interquartile range) and compared using Mann-Whitney *U* test for continuous variables, and presented as number (percentage) and compared using $\chi^2$ test for categorical variables unless otherwise specified.

*Other cancers included gynecological cancer, thyroid cancer, oral cancer, head and neck cancer, sarcoma, and carcinoma of unknown primary.

†These variables were compared using Fisher's exact test instead of $\chi^2$ test.

AF, atrial fibrillation.

### Cardiology involvement and cardiovascular and non-cardiovascular death, and stroke events in adult patients with advanced solid cancer and AF

During the follow-up period (median 1.41 years), 84 patients died, with eight deaths due to cardiovascular causes and 65 due to cancer (S3 Table). Aalen-Johansen estimates for the secondary endpoints, accounting for competing risks, showed that the cumulative incidence for neither cardiovascular nor non-cardiovascular death differed significantly between patients with and without cardiology involvement (p = 0.264 and p = 0.151, respectively; Fig 5). In addition, no patient in either of the two groups had incident cerebral infarction or hemorrhage.

## Discussion

In this retrospective cohort study, we demonstrated that comorbid AF *per se* was not independently associated with mortality in adult patients with advanced solid cancer. Furthermore, in patients with advanced cancer complicated by AF, cardiology involvement was

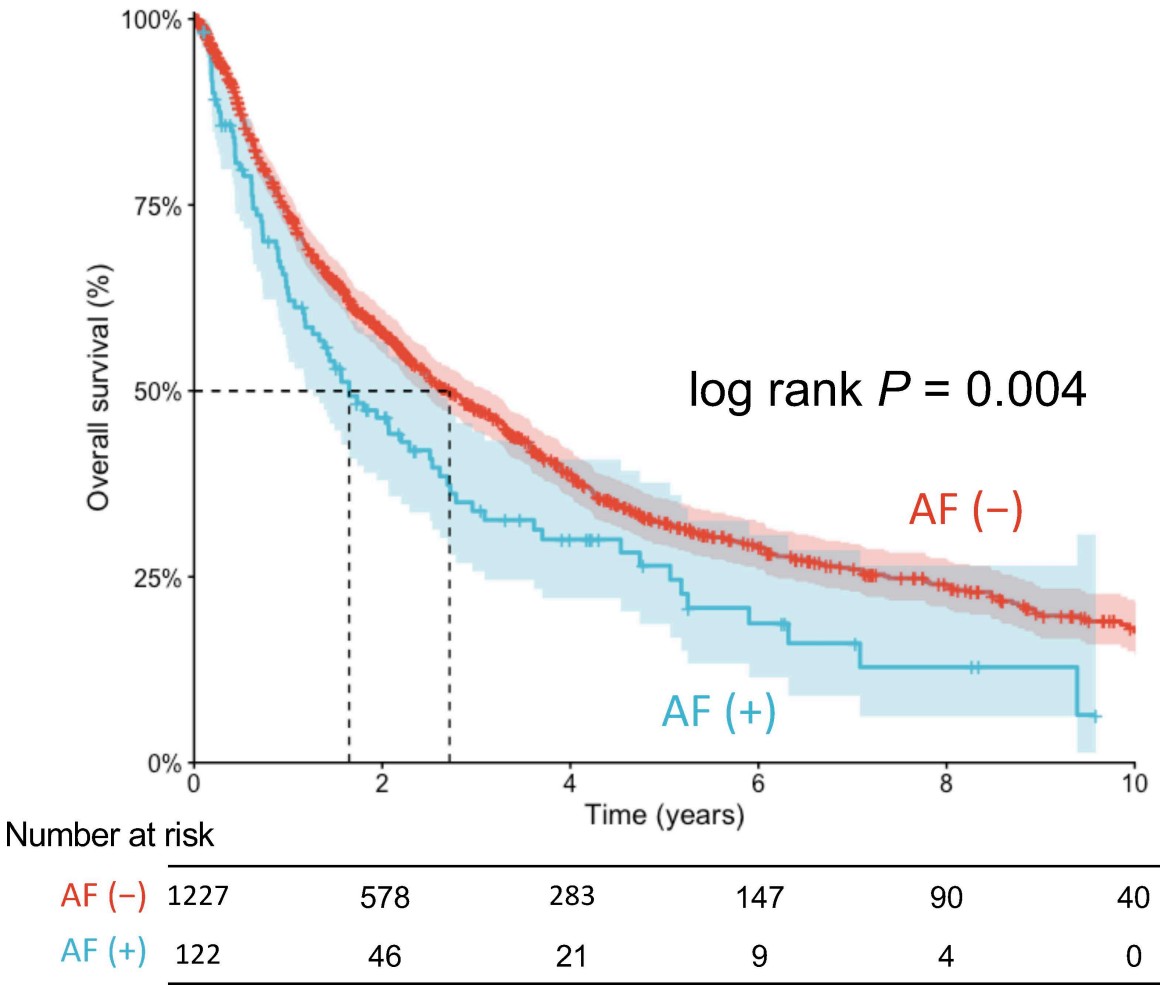

**Fig 3. Kaplan-Meier estimates of overall survival in adult patients with advanced solid cancer with and without AF.** Shaded zones indicate 95% confidence intervals. AF, atrial fibrillation.

associated with lower all-cause mortality, which was contributed by both cardiovascular and non-cardiovascular causes. These results underscore the importance of a cardio-oncology multidisciplinary approach in such patients.

We observed different risks of AF across various cancer types in patients with advanced solid cancers. The risk of comorbid AF was the highest among the patients with lung and mediastinal cancer, followed by gastrointestinal cancer, and was the lowest for breast cancer. This pattern is highly consistent with previous studies on patients with cancer at all stages [4,15], and those with metastatic cancer [16]. The increased risk of AF in patients with cancer compared to the general population can be attributed to multiple cancer-related factors, including direct locoregional effects, cardiotoxicity of anti-cancer drugs, thoracic radiotherapy, surgery, and hormonal alterations [17]. Direct tumor growth, radiotherapy to the chest, and thoracic surgery can contribute to the high risk of AF in patients with lung and mediastinal cancer [4].

One major finding of this study was that in patients with advanced solid cancer, the association between comorbid AF and increased mortality disappeared after multivariable adjustment, indicating that AF *per se* does not increase mortality in patients with advanced solid

**Table 2. The Cox proportional hazard analyses for the association between all-cause mortality and AF and other clinical characteristics.**

| | Crude | | | Multivariable | | |
|---|---|---|---|---|---|---|
| | HR | 95%CI | P-value | HR | 95%CI | P-value |
| **AF** | 1.39 | 1.11–1.75 | 0.004 | 1.08 | 0.84–1.39 | 0.552 |
| **Women** | 0.75 | 0.63–0.82 | <0.001 | 0.94 | 0.80–1.11 | 0.462 |
| **Age > 65** | 1.51 | 1.31–1.73 | <0.001 | 1.37 | 1.18–1.59 | <0.001 |
| **Hemoglobin (per 1 g/dL increase)** | 0.92 | 0.90–0.95 | <0.001 | 0.93 | 0.90–0.96 | <0.001 |
| **Creatinine (per 1 mg/dL increase)** | 1.10 | 1.01–1.20 | 0.022 | 1.08 | 0.98–1.19 | 0.184 |
| **Comorbidities** | | | | | | |
| Hypertension | 1.31 | 1.13–1.51 | <0.001 | 1.01 | 0.85–1.20 | 0.921 |
| Diabetes mellitus | 1.30 | 1.11–1.53 | 0.002 | 0.98 | 0.82–1.17 | 0.819 |
| Heart failure | 1.19 | 0.93–1.52 | 0.158 | 0.87 | 0.66–1.15 | 0.341 |
| Stroke | 1.39 | 1.09–1.78 | 0.008 | 1.00 | 0.77–1.30 | 0.975 |
| **Neoplasm type** | | | | | | |
| Urologic | 1 | – | – | 1 | – | – |
| Breast | 0.93 | 0.71–1.23 | 0.622 | 0.91 | 0.64–1.28 | 0.578 |
| Gastrointestinal | 2.54 | 1.97–3.27 | <0.001 | 2.70 | 1.96–3.73 | <0.001 |
| Lung and mediastinal | 3.61 | 2.79–4.68 | <0.001 | 2.91 | 2.10–4.05 | <0.001 |
| Hepato-biliary-pancreatic | 6.29 | 4.77–8.30 | <0.001 | 6.16 | 4.38–8.67 | <0.001 |
| Other cancers[*] | 1.30 | 0.89–1.89 | 0.175 | 1.90 | 1.24–2.90 | 0.003 |
| **Treatment for neoplasm** | | | | | | |
| Surgery | 0.71 | 0.61–0.82 | <0.001 | 0.49 | 0.41–0.59 | <0.001 |
| Radiotherapy | 1.38 | 1.13–1.69 | 0.002 | 1.32 | 1.06–1.63 | 0.012 |
| Chemotherapy | 1.62 | 1.36–1.91 | <0.001 | 0.81 | 0.64–1.02 | 0.067 |
| Endocrine therapy | 0.37 | 0.31–0.46 | <0.001 | 0.64 | 0.48–0.85 | 0.002 |

[*]Other cancers included gynecological cancer, thyroid cancer, oral cancer, head and neck cancer, sarcoma, and carcinoma of unknown primary.

AF, atrial fibrillation; CI, confidence interval; HR, hazard ratio.

cancer, and that survival of such patients is mostly determined by clinical features such as age and anemia, and oncologic features. This was further evidenced by the cancer type-stratified analysis (S2 Fig). Additionally, the presence of heart failure was not associated with mortality, either, further supporting that age, anemia, and oncologic features may have larger impact than cardiovascular comorbidities in such patients with advanced solid cancer. Several previous studies have reported the negative impact of comorbid AF on morbidity and mortality [6,18,19]. Hussian et al. identified AF as an independent predictor of all-cause mortality after multivariable adjustment in patients with cancer [18]. The discrepancy between our findings and those of Hussain et al. may stem from our inclusion criteria, which focused exclusively on patients with advanced cancer; however, Hussain et al. included patients with cancer of all stages. Additionally, the different covariates used in the multivariable analyses may have contributed to the differing results [18]. Further studies with larger sample sizes are warranted to elucidate the role of comorbid AF in the prognosis of patients with cancer, taking cancer stages and types into consideration.

Unexpectedly, among the 122 patients with cancer and AF in this study, 42% did not receive care from a cardiology provider, despite the European Society for Medical Oncology's recommendation that patients with cancer and AF should receive appropriate medical care as indicated (Grade of recommendation C, Level of evidence IV) [20]. Furthermore, we observed that among patients with advanced solid cancer complicated with AF, cardiology involvement in patient care was associated with lower all-cause mortality. Few studies have focused on the

**Table 3. Baseline characteristics of the participants with AF and the comparison between those with and without cardiology involvement.**

| | Total patients with AF (N = 122) | No Cardiologist (N = 51) | With Cardiologist (N = 71) | P-value |
|---|---|---|---|---|
| Age, years | 74 (68–80) | 74 (64–81) | 74 (70–80) | 0.858 |
| Women | 33 (27.0) | 14 (27.5) | 19 (26.8) | 1.000 |
| Hemoglobin (g/dL) | 12.0 (10.1–13.9) | 12.3 (10.4–14.2) | 11.9 (10.2–13.6) | 0.159 |
| Creatinine (mg/dL) | 0.81 (0.68–1.00) | 0.79 (0.69–1.00) | 0.82 (0.68–0.99) | 0.791 |
| **Comorbidities** | | | | |
| Hypertension | 72 (59.0) | 23 (45.1) | 49 (69.0) | 0.014 |
| Diabetes mellitus | 42 (34.4) | 16 (31.4) | 26 (36.6) | 0.683 |
| Heart failure | 43 (35.2) | 9 (17.6) | 34 (47.9) | 0.001 |
| Stroke | 16 (13.1) | 10 (19.6) | 6 (8.5) | 0.126 |
| **Classification of AF** | | | | 0.264 |
| Paroxysmal | 73 (59.8) | 34 (66.7) | 39 (54.9) | |
| Persistent | 49 (40.2) | 17 (33.3) | 32 (45.1) | |
| **CHADS2 score** | | | | 0.353 |
| 0 ~ 1 | 43 (35.2) | 21 (41.2) | 22 (31.0) | |
| 2 ~ 3 | 62 (50.8) | 22 (43.1) | 40 (56.3) | |
| 4 ~ 6 | 17 (13.9) | 8 (15.7) | 9 (12.7) | |
| **Neoplasm type** | | | | |
| Gastrointestinal | 41 (33.6) | 17 (33.3) | 24 (33.8) | 1.000 |
| Urologic | 18 (14.8) | 6 (11.8) | 12 (16.9) | 0.596 |
| Hepato-biliary-pancreatic | 9 (7.4) | 2 (3.9) | 7 (9.9) | 0.302[†] |
| Breast | 7 (5.7) | 1 (2.0) | 6 (8.5) | 0.237[†] |
| Lung and mediastinal | 39 (32.0) | 21 (41.2) | 18 (25.4) | 0.099 |
| Other cancers[*] | 8 (6.6) | 4 (7.8) | 4 (5.6) | 0.718[†] |
| **Staging of neoplasm** | | | | 0.639[†] |
| Stage II ~ III | 4 (3.3) | 1 (2.0) | 3 (4.2) | |
| Stage IV | 1307 (96.7) | 50 (98.0) | 68 (95.8) | |
| **Treatment for neoplasm** | | | | |
| Surgery | 46 (37.7) | 18 (35.3) | 28 (39.4) | 0.782 |
| Radiotherapy | 19 (15.6) | 12 (23.5) | 7 (9.9) | 0.072 |
| Chemotherapy | 77 (63.1) | 32 (62.7) | 45 (63.4) | 1.000 |
| Endocrine therapy | 18 (14.8) | 4 (7.8) | 14 (19.7) | 0.076[†] |
| **Treatment for AF** | | | | |
| Anticoagulants | 62 (50.8) | 11 (21.6) | 51 (71.8) | <0.001 |
| Antiarrhythmics | 85 (69.7) | 31 (60.8) | 54 (76.1) | 0.107 |
| **Vaughan Williams classification** | | | | |
| I | 14 (11.5) | 4 (7.8) | 10 (14.1) | 0.391[†] |
| II (beta-blockers) | 55 (45.1) | 15 (29.4) | 40 (56.3) | 0.003 |
| III | 6 (4.9) | 0 (0) | 6 (8.5) | 0.040[†] |
| IV | 42 (34.4) | 19 (37.3) | 23 (32.4) | 0.716 |

Data are presented as median (interquartile range) and compared using Mann-Whitney $U$ test for continuous variables, and as number (percentage) and compared using $\chi^2$ test for categorical variables unless otherwise specified.

[*]Other cancers included gynecological cancer, thyroid cancer, oral cancer, head and neck cancer, sarcoma, and carcinoma of unknown primary.

[†]These variables were compared using Fisher's exact test instead of $\chi^2$ test.

AF, atrial fibrillation.

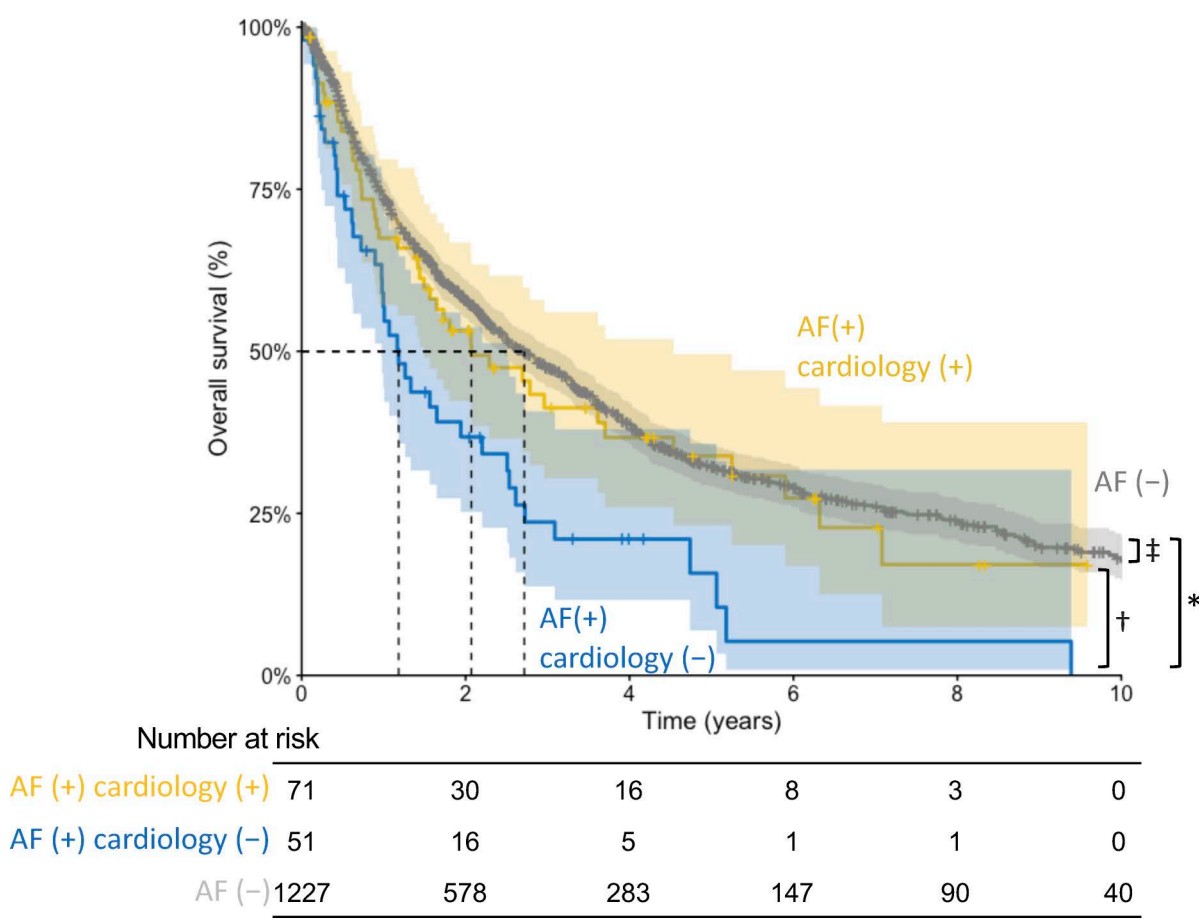

**Fig 4. Kaplan-Meier estimates of overall survival among adult patients with advanced solid cancer without AF, with AF and cardiology involvement, and with AF without cardiology involvement.** Shaded zones indicate 95% confidence intervals. *Log-rank **p** < 0.001. †Log-rank **p** = 0.016. ‡Log-rank **p** = 0.475. AF, atrial fibrillation.

prognostic effect of regular cardiologist participation in the care of patients with cancer complicated by AF. O'Neal et al. previously demonstrated that patients with cancer complicated by AF are more likely to fill prescriptions for anticoagulants if treated by a cardiologist and that cardiology involvement in such patients is associated with reduced stroke risk without increased bleeding risk [11]. In this study, we also noted that patients under cardiology care were more likely to take anticoagulants and antiarrhythmics, particularly beta-blockers. However, among the patients with advanced cancer, no patient had incident cerebral infarction or hemorrhage, and the cause of death was predominantly cancer, with only a small number of cardiovascular deaths. Both cardiovascular and non-cardiovascular mortality rates were numerically lower, but without statistical significance, in those with cardiology involvement compared to those without cardiology involvement. The limited number of patients and events, leading to the limited power of analysis, may have contributed to the lack of significant difference. The mechanism behind the reduced mortality, though remains unclear and warrants further investigation, could possibility be attributed to the comprehensive evaluation of patients prompted by the multidisciplinary approach involving cardiologists. The results of this study further support the guidelines statement that patients with both advanced solid

**Table 4. The crude and adjusted hazard ratios of cardiology involvement and clinical characteristics for all-cause mortality in patients with AF and advanced solid cancer.**

| | Crude | | | Model 1 | | | Model 2 | | | Model 3 | | |
|---|---|---|---|---|---|---|---|---|---|---|---|---|
| | HR | 95%CI | P-value | HR | 95%CI | P-value | HR | 95%CI | P-value | HR | 95%CI | P-value |
| **Cardiology involvement** | 0.59 | 0.38–0.91 | 0.017 | 0.50 | 0.31–0.82 | 0.006 | 0.51 | 0.31–0.84 | 0.009 | 0.50 | 0.28–0.88 | 0.017 |
| **Women** | 0.86 | 0.52–1.40 | 0.541 | 0.84 | 0.50–1.41 | 0.507 | 0.79 | 0.44–1.43 | 0.440 | 0.70 | 0.38–1.29 | 0.257 |
| **Age > 65** | 0.93 | 0.56–1.52 | 0.762 | 1.20 | 0.66–2.16 | 0.553 | 1.56 | 0.80–3.02 | 0.190 | 1.57 | 0.77–3.19 | 0.211 |
| **Persistent AF** | 0.91 | 0.58–1.42 | 0.677 | 0.94 | 0.59–1.49 | 0.791 | 1.16 | 0.72–1.87 | 0.547 | 1.34 | 0.79–2.27 | 0.280 |
| **Hemoglobin (per 1 g/dL increase)** | 0.93 | 0.86–1.02 | 0.121 | 0.90 | 0.82–0.99 | 0.023 | 0.90 | 0.82–0.99 | 0.029 | 0.91 | 0.82–1.00 | 0.043 |
| **Creatinine (per 1 mg/dL increase)** | 0.96 | 0.75–1.21 | 0.693 | 0.99 | 0.77–1.28 | 0.963 | 1.04 | 0.79–1.37 | 0.783 | 1.07 | 0.79–1.44 | 0.679 |
| **Comorbidities** | | | | | | | | | | | | |
| Hypertension | 0.88 | 0.57–1.37 | 0.569 | 0.85 | 0.50–1.44 | 0.545 | 0.87 | 0.49–1.53 | 0.621 | 0.78 | 0.43–1.40 | 0.404 |
| Diabetes mellitus | 1.21 | 0.77–1.90 | 0.409 | 1.42 | 0.85–2.38 | 0.178 | 1.43 | 0.83–2.48 | 0.201 | 1.44 | 0.83–2.48 | 0.192 |
| Heart failure | 0.88 | 0.56–1.37 | 0.563 | 0.89 | 0.53–1.49 | 0.667 | 0.99 | 0.56–1.72 | 0.959 | 1.19 | 0.65–2.18 | 0.562 |
| Stroke | 1.17 | 0.62–2.23 | 0.623 | 1.04 | 0.53–2.05 | 0.915 | 0.76 | 0.36–1.58 | 0.460 | 0.74 | 0.34–1.59 | 0.435 |
| **Neoplasm type** | | | | | | | | | | | | |
| Urologic | 1.00 | – | – | | | | 1.00 | – | – | 1.00 | – | – |
| Breast | 1.12 | 0.32–3.86 | 0.858 | | | | 1.79 | 0.43–7.40 | 0.422 | 2.25 | 0.51–9.90 | 0.283 |
| Gastrointestinal | 2.64 | 1.12–6.20 | 0.026 | | | | 2.57 | 1.05–6.27 | 0.039 | 3.06 | 1.22–7.69 | 0.017 |
| Lung and mediastinal | 5.07 | 2.20–11.69 | <0.001 | | | | 5.85 | 2.39–14.32 | <0.001 | 6.73 | 2.70–16.82 | <0.001 |
| Hepato-biliary-pancreatic | 11.17 | 3.90–31.97 | <0.001 | | | | 10.48 | 3.57–30.79 | <0.001 | 13.61 | 4.20–44.19 | <0.001 |
| Other cancers* | 4.97 | 1.70–14.48 | 0.003 | | | | 7.89 | 2.46–25.26 | <0.001 | 11.17 | 3.27–38.15 | <0.001 |
| **Treatment for AF** | | | | | | | | | | | | |
| Beta-blocker | 0.64 | 0.41–1.00 | 0.052 | | | | | | | 0.91 | 0.49–1.68 | 0.753 |
| Antiarrhythmics except beta-blocker | 1.15 | 0.75–1.77 | 0.522 | | | | | | | 1.72 | 1.01–2.91 | 0.044 |
| Anticoagulants | 0.62 | 0.40–0.96 | 0.031 | | | | | | | 0.96 | 0.51–1.82 | 0.905 |

Different models include cardiology involvement and covariables as listed in the table. Briefly, model 1 was adjusted for age, sex, AF classification, and comorbidities. Model 2 was further adjusted for cancer types. In Model 3, additional adjustments were made for treatments specific to AF.

*Other cancers included gynecological cancer, thyroid cancer, oral cancer, head and neck cancer, sarcoma, and carcinoma of unknown primary.

AF, atrial fibrillation; CI, confidence interval; HR, hazard ratio.

cancer and AF should be comprehensively evaluated for cardiovascular risks and drug-drug interactions, through a multidisciplinary cardio-oncology approach [11].

## Study limitations

This study had several limitations. First, this was a retrospective cohort study conducted at a single center, and the total number of patients with advanced solid cancer complicated by AF was limited, making the findings of this study hypothesis-generating. Patients who were lost to follow-up might have resulted in potential selection bias. Second, given the nature of observational studies, the observed association between cardiology involvement and mortality in patients with advanced cancer and AF may have been biased by unobserved confounders, including confounding by indication. Similarly, such unobserved confounders, as well as competing risks might have also influenced the analysis for the association between AF and mortality. Third, detailed information on specific anti-malignancy medicines such as anthracyclines, trastsuzumab and immune checkpoint inhibitors was not available in regards with cardiotoxicity. Finally, this study included only patients with advanced solid cancers; therefore, the results were not generalizable to those with early-stage cancers or hematological malignancies.

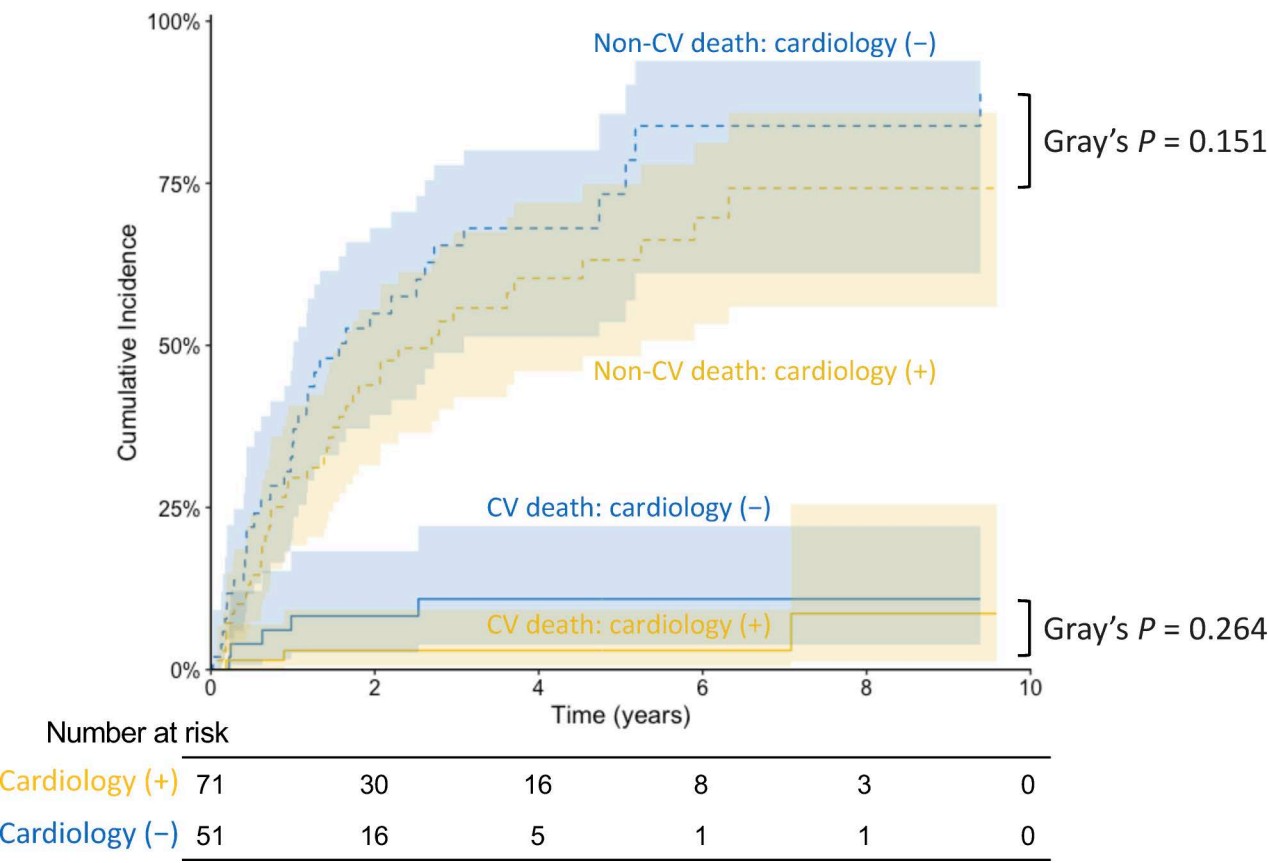

**Fig 5. Cumulative incidences of cardiovascular and non-cardiovascular deaths stratified by cardiology involvement among adult patients with advanced solid cancer and AF. Solid lines indicate cumulative incidences of cardiovascular deaths; and dashed lines indicate cumulative incidences of non-cardiovascular deaths. Shaded zones indicate 95% confidence intervals. AF, atrial fibrillation; CV, cardiovascular.**

## Conclusions

In adult patients with advanced solid cancer, AF *per se* was not independently associated with increased mortality. In patients with advanced solid cancer complicated by AF, involvement of a cardiology provider was associated with lower all-cause mortality, likely contributed by both cardiovascular and non-cardiovascular causes, but with low certainty that this finding is not attributable to unmeasured confounding. The findings underscore the importance of multi-disciplinary cardio-oncologic collaboration in patients with cancer, even in advanced stages, when complicated by AF, and warrant further investigation in large prospective cohorts.

## Supporting information

**S1 Fig. Scaled Schoenfeld residual plots for variables included in the multivariable Cox proportional hazards model corresponding to the results in** <u>Table 1</u>**.** P values for all variables including the presence of AF, as well as that of the global Schoenfeld test were > 0.05, indicating that the proportional hazard assumptions were met. HF, heart failure. (Tiff)

**S2 Fig. Kaplan-Meier estimates for overall survival in adult patients with advanced solid cancer with and without AF, stratified by cancer type.** Patients with advanced urologic,

gastrointestinal, and lung and mediastinal (lung and med) cancers were analyzed. Shaded zones indicate 95% confidence intervals. AF, atrial fibrillation.
(Tiff)

**S3 Fig. Scaled Schoenfeld residual plots for variables included in the multivariable Cox proportional hazards models corresponding to the results in** Table 4. Panels A, B and C correspond to the scaled Schoenfeld residual plots for variables included in Models 1, 2 and 3 in Table 4, respectively. P value for each variable, as well as that for the global Schoenfeld test were > 0.05 in all models, indicating that the proportional hazard assumptions were met. AA, antiarrhythmics; AF, atrial fibrillation; HF, heart failure.
(Tiff)

**S1 Table. Different treatment strategies among different cancer types in the entire cohort.**
(DOCX)

**S2 Table. Different treatment strategies among different cancer types in patient with atrial fibrillation.**
(DOCX)

**S3 Table. Causes of mortality in patients with AF and advanced solid cancer.**
(DOCX)

## Author contributions

**Conceptualization:** Takeshi Sato, Zhehao Dai, Jun Hashimoto, Nobuyuki Komiyama, Takayuki Inomata, Teruo Yamauchi.

**Data curation:** Takeshi Sato, Zhehao Dai.

**Formal analysis:** Takeshi Sato, Zhehao Dai.

**Methodology:** Takeshi Sato, Zhehao Dai, Jun Hashimoto, Sachiko Ohde.

**Project administration:** Jun Hashimoto.

**Supervision:** Jun Hashimoto, Sachiko Ohde, Nobuyuki Komiyama, Teruo Yamauchi.

**Validation:** Jun Hashimoto, Sachiko Ohde, Nobuyuki Komiyama, Takayuki Inomata.

**Visualization:** Zhehao Dai.

**Writing – original draft:** Takeshi Sato, Zhehao Dai.

**Writing – review & editing:** Jun Hashimoto, Sachiko Ohde, Nobuyuki Komiyama, Takayuki Inomata, Teruo Yamauchi.

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
