## [Decision Letter · Decision Letter 0]

29 Dec 2024

PONE-D-24-53250Cardiology Involvement and Mortality in Adult Patients with Advanced Solid Cancer Complicated by Atrial FibrillationPLOS ONE

Dear Dr. Dai,

Thank you for submitting your manuscript to PLOS ONE. After careful consideration, we feel that it has merit but does not fully meet PLOS ONE’s publication criteria as it currently stands. Therefore, we invite you to submit a revised version of the manuscript that addresses the points raised during the review process.

We look forward to receiving your revised manuscript.

Kind regards,

Yoshihiro Fukumoto

Academic Editor

PLOS ONE

Journal Requirements: When submitting your revision, we need you to address these additional requirements. 1. Please ensure that your manuscript meets PLOS ONE's style requirements, including those for file naming. The PLOS ONE style templates can be found at https://journals.plos.org/plosone/s/file?id=wjVg/PLOSOne_formatting_sample_main_body.pdf and https://journals.plos.org/plosone/s/file?id=ba62/PLOSOne_formatting_sample_title_authors_affiliations.pdf 2. In the online submission form, you indicated that "Data cannot be shared publicly because of restrictions from the institutional review board. Instead, anonymized data are available upon reasonable requests to the corresponding author (Zhehao Dai, daizh@luke.ac.jp) with confidentiality agreements assured." All PLOS journals now require all data underlying the findings described in their manuscript to be freely available to other researchers, either 1. In a public repository, 2. Within the manuscript itself, or 3. Uploaded as supplementary information.This policy applies to all data except where public deposition would breach compliance with the protocol approved by your research ethics board. If your data cannot be made publicly available for ethical or legal reasons (e.g., public availability would compromise patient privacy), please explain your reasons on resubmission and your exemption request will be escalated for approval. 3. Please ensure that you refer to Figure 5 in your text as, if accepted, production will need this reference to link the reader to the figure. 4. Please include captions for your Supporting Information files at the end of your manuscript, and update any in-text citations to match accordingly. Please see our Supporting Information guidelines for more information: http://journals.plos.org/plosone/s/supporting-information.

Reviewers' comments:

Reviewer's Responses to Questions

**Comments to the Author**

1. Is the manuscript technically sound, and do the data support the conclusions?

Reviewer #1: Yes

Reviewer #2: Partly

2. Has the statistical analysis been performed appropriately and rigorously? 

Reviewer #1: Yes

Reviewer #2: Yes

3. Have the authors made all data underlying the findings in their manuscript fully available?

Reviewer #1: Yes

Reviewer #2: Yes

4. Is the manuscript presented in an intelligible fashion and written in standard English?

Reviewer #1: Yes

Reviewer #2: Yes

5. Review Comments to the Author

Reviewer #1: This is an interesting paper that retrospectively evaluated the prognostic impact of Af in solid cancers at a single center.

1.Heart failure as a comorbidity did not show any effect on prognosis. Is this related to the fact that atrial fibrillation did not show any effect on prognosis after adjustment for factors in patients with solid cancer?

2.Could the involvement of the cardiology department have been a bias? For example, cancer patients with poor prognosis were not referred.

Reviewer #2: Thank you for the opportunity to review “Cardiology Involvement and Mortality in Adult Patients with Advanced Solid Cancer Complicated by Atrial Fibrillation”

The authors investigated the association between comorbid atrial fibrillation (AF) and survival in adult patients with advanced solid cancer and the impact of cardiology involvement in such patients. This study was a meaningful with important knowledge, and a well-performed analysis. However, I have a few concerns about this manuscript.

The authors showed that in patients with advanced solid cancer and atrial fibrillation, cardiology involvement is associated with reduced overall mortality. This is a very important finding, but what is the reason of this? Specifically, you showed that the cumulative incidence for neither cardiovascular nor non-cardiovascular death differed significantly between patients with and without cardiology involvement.

Specific comment:

1) The stage of the cancer would be related to the prognosis. Also, In the case of worse the cancer prognosis, it is possible that oncologists are less likely to consult with cardiologists, leading to the less chance to be involved by cardiologists.

Please provide the information of the stage of cancer between patients with and without cardiology involvement at the baseline in this study.

2) Comorbidities such as severe anemia and renal failure may be related to the prognosis of cancer patients. Please provide data of hemoglobin and creatine at the baseline. We also suggest additionally adjust for anemia and renal failure in the Cox proportional hazard model.

3) Authors showed that anticoagulants were frequently administered to patients under cardiology care. What is the incidence of cerebrovascular disease (cerebral infarction and/or cerebral hemorrhage) between the two groups? Similarly, is there any difference of anemia between the two groups? It is expected that the presence of anemia will make it difficult to introduce anticoagulant therapy.

4) Is there a difference in prognosis between groups treated for AF and those not treated for AF among the patient with cardiologist involvement?

5) The figure numbers seem to differ from those in the manuscript (Figures 2-5).

6. PLOS authors have the option to publish the peer review history of their article (what does this mean? ). If published, this will include your full peer review and any attached files.

**Do you want your identity to be public for this peer review?** For information about this choice, including consent withdrawal, please see our Privacy Policy .

Reviewer #1: No

Reviewer #2: No

---

## [Author Response · Author response to Decision Letter 0]

8 Jan 2025

We have uploaded our response to reviewers as a seperate file, named "Response to reviewers.docx".

---

## [Decision Letter · Decision Letter 1]

31 Jan 2025

Cardiology involvement and mortality in adult patients with advanced solid cancer complicated by atrial fibrillation

PONE-D-24-53250R1

Dear Dr. Dai,

We’re pleased to inform you that your manuscript has been judged scientifically suitable for publication and will be formally accepted for publication once it meets all outstanding technical requirements.

Kind regards,

Yoshihiro Fukumoto

Academic Editor

PLOS ONE

Additional Editor Comments (optional):

Reviewers' comments:

Reviewer's Responses to Questions

**Comments to the Author**

1. If the authors have adequately addressed your comments raised in a previous round of review and you feel that this manuscript is now acceptable for publication, you may indicate that here to bypass the “Comments to the Author” section, enter your conflict of interest statement in the “Confidential to Editor” section, and submit your "Accept" recommendation.

Reviewer #1: All comments have been addressed

Reviewer #2: All comments have been addressed

2. Is the manuscript technically sound, and do the data support the conclusions?

Reviewer #1: Yes

Reviewer #2: Yes

3. Has the statistical analysis been performed appropriately and rigorously? 

Reviewer #1: Yes

Reviewer #2: Yes

4. Have the authors made all data underlying the findings in their manuscript fully available?

Reviewer #1: Yes

Reviewer #2: Yes

5. Is the manuscript presented in an intelligible fashion and written in standard English?

Reviewer #1: Yes

Reviewer #2: Yes

6. Review Comments to the Author

Reviewer #1: I think the content of this paper is excellent. The resubmitted paper has been appropriately revised to address the issues pointed out. In conclusion, there are no further points to be made about this paper.

Reviewer #2: Thank you very much for revising the original submission.

The authors answered the questions from the reviewers nicely in this R1 session.

The reviewer would like to congratulate the authors for the current interesting study.

7. PLOS authors have the option to publish the peer review history of their article (what does this mean? ). If published, this will include your full peer review and any attached files.

**Do you want your identity to be public for this peer review?** For information about this choice, including consent withdrawal, please see our Privacy Policy .

Reviewer #1: No

Reviewer #2: No

---

## [Editor Report · Acceptance letter]

PONE-D-24-53250R1

PLOS ONE

Dear Dr. Dai,

I'm pleased to inform you that your manuscript has been deemed suitable for publication in PLOS ONE. Congratulations! Your manuscript is now being handed over to our production team.

Kind regards,

on behalf of

Dr. Yoshihiro Fukumoto

Academic Editor

PLOS ONE